# Research on the Rapid Diagnostic Method of Rolling Bearing Fault Based on Cloud–Edge Collaboration

**DOI:** 10.3390/e24091277

**Published:** 2022-09-10

**Authors:** Xianghong Tang, Lei Xu, Gongsheng Chen

**Affiliations:** 1State Key Laboratory of Public Big Data, College of Computer Science and Technology, Guizhou University, Guiyang 550025, China; 2Key Laboratory of Advanced Manufacturing Technology of the Ministry of Education, Guizhou University, Guiyang 550025, China

**Keywords:** fast diagnosis, depth-separable convolution, transfer learning, information fusion, data security

## Abstract

Recent deep-learning methods for fault diagnosis of rolling bearings need a significant amount of computing time and resources. Most of them cannot meet the requirements of real-time fault diagnosis of rolling bearings under the cloud computing framework. This paper proposes a quick cloud–edge collaborative bearing fault diagnostic method based on the tradeoff between the advantages and disadvantages of cloud and edge computing. First, a collaborative cloud-based framework and an improved DSCNN–GAP algorithm are suggested to build a general model using the public bearing fault dataset. Second, the general model is distributed to each edge node, and a limited number of unique fault samples acquired by each edge node are used to quickly adjust the parameters of the model before running diagnostic tests. Finally, a fusion result is made from the diagnostic results of each edge node by DS evidence theory. Experiment results show that the proposed method not only improves diagnostic accuracy by DSCNN–GAP and fusion of multi-sensors, but also decreases diagnosis time by migration learning with the cloud–edge collaborative framework. Additionally, the method can effectively enhance data security and privacy protection.

## 1. Introduction

The functioning status of the rolling bearing, a frequently utilized fundamental component, has a direct impact on the normal operation of the rotating mechanism and even the entire piece of machinery. It plays a crucial role in mechanical equipment and is typically the component that fails the most frequently. Rolling bearing failures in rotating machinery account for 40% of mechanical failures, according to statistics on mechanical failures [1,2]. Therefore, research on bearing health monitoring and fault diagnostics is crucial to ensuring the safe and reliable functioning of mechanical equipment.

The algorithms of bearing fault diagnosis are constantly being improved. The equipment collects time series vibration signals, which are then sent to the cloud for unified processing and analysis. In terms of algorithms, the wavelet transform [3], envelope analysis [4], the short-time Fourier transform [5], and other diagnostic techniques are frequently utilized in the cloud. Most of these are theoretically interpretable, but the bearing vibration signal’s nonlinearity and non-stationarity faults make diagnostic accuracy poor. Compared to conventional signal processing techniques, algorithms such as SVM [6], Naive Bayes [7], K-closest neighbors [8], etc., have significantly increased diagnostic accuracy and usability. These techniques, however, are highly dependent on the accuracy of artificial feature extraction, which depends on professional judgment and past knowledge. Some researchers use deep learning technology, such as Convolutional Neural Networks (CNNs) [9], Deep Belief Networks (DBNs) [10], Sparse Auto-encoders (SAE) [11], etc., in the field of bearing failure diagnostics in order to solve the aforementioned issues and accomplish feature self-extraction. Qiao [12] proposed a novel weighted multi-scale convolutional neural network for adaptive multi-scale fusion of features from the original vibration signal. Zhang [13] proposed a transfer learning method based on CNNs for fault diagnosis in order to utilize data under different working conditions to improve performance. Wen [14] used transfer learning to improve the CNN structure for fault diagnosis. In order to correlate and coordinate data from many information sources, multi-sensing data that takes advantage of the advancements in sensor technology can more precisely depict the state of equipment health. Yean [15] combined the results of multisource information fusion which can effectively describe the fault type. Chao [16] et al. proposed that the multi-sensor data fusion method greatly improves the accuracy and robustness in the fault diagnosis of axial piston pumps. As a way to increase target trust evaluation, many scholars have used the DS (Dempster–Shafer) evidence theory algorithm to obtain better diagnostic performance. Pei [17] et al. proposed a rolling bearing fault diagnostic method based on DS evidence theory fusion. DS evidence theory is used to fuse multi-sensor data. Sun [18] proposed an improved DS evidence theory, which can produce acceptable outcomes by reassigning the weighting factors prior to fusion. Due to the high complexity of the algorithm, the fault diagnostic model based on deep learning needs a lot of computing resources and time, and the diagnostic model based on the edge computing framework has emerged as a successful method to address the issue.

The cloud computing-based fault diagnostic system has a lot of flaws. The data gathered on the device side must be sent to the cloud if the model training is only carried out in the cloud. Uploading the data to the cloud computing facility will enhance the danger of privacy leaks because the data generated on the device often contains the user’s private information [19,20]. Additionally, there is a possibility of high-latency transmission during data transmission from the device to the cloud computing facility [21]. Furthermore, the deep learning fault diagnostic model needs to be thoroughly trained on a substantial amount of labeled sample training data in order to successfully accomplish correct fault identification [22,23]. The collecting of labeled data and exact training both take a significant amount of time and processing power. With edge computing, the issue of privacy leakage is avoided because user data is processed and stored locally on edge computing nodes. Edge computing distributes applications, processing, and storage across edge devices and cloud data centers at the same time. Attacks by hackers only affect the local data on a particular device or edge node, not the entire data set. Edge computing therefore possesses the traits of high security [24]. The edge computing framework uses the benefit of the edge node being close to the user in order to tackle the issue of high latency produced by the data uploaded to the cloud by the device side, and reduces the issue of network delay while increasing processing efficiency. Qian [25] et al. proposed an edge computing-based method for real-time fault diagnosis and dynamic control of rotating machines which reduced the system response time, transmission bandwidth occupation, and storage and computation resources on the cloud. Wang [26] et al. implemented an edge computing-based permanent-magnet synchronous motor-bearing fault diagnosis which made the execution of a complicated algorithm become practical. Qu [27] et al. demonstrated one can quickly and effectively diagnose equipment defects through the use of edge computing in industrial processing and production, which raises the system operating efficiency of the controlling system. Considering the problem that model training is difficult due to the very limited computing and storage capacity of the edge end in the edge computing framework [25,28], this work integrates the deep learning concept of merging cloud resources with high-performance computing and storage, as well as edge devices with strong individual adaptability and well-controlled time limitations, based on the research on cross-domain learning. It suggests the use of a bearing defect diagnostic approach for cloud and edge collaboration. This study develops a method for diagnosing and analyzing bearing faults based on intelligent edge and cloud–edge collaboration. The main contributions of this paper are as follows:(1)This paper designs an overall fault diagnostic framework that aims to coordinate the computing resources of the cloud edge and meet the purpose of real-time intelligent diagnosis of bearings. It does this by combining the benefits of abundant cloud computing resources, large storage capacity, and good real-time performance of edge computing.(2)This study suggests an enhanced deep learning lightweight method based on DSCNN–GAP that enhances the timeliness and accuracy of fault diagnosis.(3)In this paper, DS evidence theory was used to create decision fusion for the diagnostic results of each edge node, and multi-sensor technology was used in this study to prevent the signal obtained from a single vibration sensor from being unable to fully monitor the running state of the bearing.(4)In this study, the transfer learning approach was used, and the deep learning ideas of fusing cloud resources and high-performance computing storage were combined. The edge nodes received the trained weights from the cloud, and experiments with the transfer of the signals from various sensors were conducted in turn.

The rest of the paper is structured as follows. The second section presents the cloud–edge collaboration diagnostic architecture and algorithm, the third introduces the DS evidence theory, and the fourth introduces the relevant experiments and results analysis. The summary and recommendations for further study are included in Section 5 of this paper.

## 2. The Cloud–Edge Collaboration Diagnostic Framework and Algorithm

### 2.1. The Cloud–Edge Collaboration Diagnostic Framework

This research, which focuses on the quick detection of bearing problems, examines the benefits and drawbacks of cloud and edge technologies and develops a cloud–edge collaborative framework for intelligent fault diagnosis of bearings, as seen in Figure 1. The framework is made up of three layers: device side, edge side, and cloud. Devices that are aware of data and sensor clusters that gather data are included in the device layer, which is at the bottom. The personalized bearing’s real-time status data is collected at the edge by the edge node using a data-sensing device, and after being processed is uploaded to the cloud for storage and analysis. Mainstream deep learning model libraries and training sample data are primarily stored in the cloud. Public bearing fault data sets, such those from Case Western Reserve University, are included in the training sample data, as well as individual bearing fault data sets that each edge node has acquired.

The specific diagnostic process and steps are as follows:The sensor cluster continuously gathers and uploads the manufacturing equipment layer’s vibration signal data to neighboring edge nodes.The edge nodes’ collection of vibration signal data creates training samples of customized conditions at each node. The data is then time-stamped, compressed, and sent to the cloud.The cloud initially trains the public defective bearing data using the improved DSCNN–GAP algorithm, and it then gradually adds personalized training data to update the global model over time.The cloud-trained universal model is downloaded by the edge nodes, which then use the training samples of customized conditions created at each edge node to transfer and learn the model. This allows the edge nodes to create customized models at each node to meet the demands of real-time bearing fault diagnosis.The DS evidence theory is applied at other nodes to perform decision fusion on each edge node’s diagnostic results and output the diagnostic results once more in order to more fully explain the running health condition of the equipment.The system will immediately issue a matching warning if the output is a certain type of defect.

### 2.2. Diagnostic Algorithm Adapted to Cloud–Edge Collaboration

In order to take into account that the diagnostic algorithm has strong applicability in both the cloud and the edge, from the perspective of a lightweight algorithm, this work presents a fault diagnosis research based on the DSCNN–GAP algorithm for cloud–edge collaboration. As seen in Figure 2, data reconstruction is used to transform the initial one-dimensional time series signal into a two-dimensional training sample for feature maps. Next, feature extraction is carried out using deep separable convolution. The precise procedure is to carry out point-by-point convolution after connection, train with separate channel characteristics, and then perform spatial convolution independently on the input data channels. After numerous iterations of convolution pooling, the fully connected layer will provide a large number of parameters and take a long time to train and test. A global average pooling layer (GAP) is created to replace the fully connected layer structure in the conventional DSCNN network in order to address the aforementioned issues. The final diagnostic result is generated directly by the softmax layer, and the global average pooling layer will automatically complete dimension transformation as well as parameter compression and reduction. The entire procedure is separated into four layers: input layer, feature extraction layer, global average pooling layer, and output layer.

The input layer is primarily used to accept the original data and carry out data normalization and standardization tasks, producing batch training samples that the deep learning model can recognize. The initial data collected by the equipment monitoring system, however, is frequently a one-dimensional time-series of vibration signals mingled with noise interference. The original time-series data are converted into grayscale picture signals for feature extraction in order to enhance the signal quality, which can greatly alleviate this issue. The data is processed in accordance with Formula (1), and the signals are converted into pixels with data between [0, 255], which can effectively prevent the influence of significant differences between signals collected by different sensors on diagnostic outcomes. This method is inspired by the composition principle of computer pixels. The limited fault data is then expanded, lowering the likelihood of over-fitting.
(1)$$P(p,n)=round\left(\frac{x_{i}-x_{\min}}{x_{\max}-x_{\min}}*255\right)$$

*P* stands for the converted pixel signal, *round* () stands for rounding, *x**_i_* stands for the data’s *i*th sample point, and *x_max_* and *x_min_* stand for the data’s maximum and minimum values, respectively. Figure 3 illustrates the data reconstruction principle. In order to reduce the impact of technician experience on diagnostic outcomes, this data processing approach can be calculated without any specified parameters while also maintaining the two-dimensional properties of the original signal.

In the entire feature extraction process, the feature extraction layer can be separated into two parts: deep convolution and point-by-point convolution. Deep convolution is the initial step of deeply separable convolution, as depicted in Figure 4. For each input channel, a different filter is used in deep convolution. The number of output feature maps is equal to the number of input channels after the deep convolution of the input channel. The deep convolution procedure can be described as follows when the total number of channels is unaltered after deep convolution:(2)$$D\text{\_}\mathrm{Conv}{\left(w,x\right)}_{\left(i,j\right)}=\sum_{m,n}^{M,N}w_{\left(m,n\right)}•x_{\left(i+m,j+n\right)}$$

Among them, *x* is the input feature map of the convolution layer, *w* is the weight matrix of the convolution kernel, (*i*, *j*) is the coordinate point of the output feature map, and *m*, *n* and *k* are the three dimensions of the convolution kernel.

The second step of deeply separable convolution is the point-by-point convolution, that is, a 1 × 1 convolution is performed on the output of the deep convolution of the first step. The point-by-point convolution can extract spatial features, will change the size of the feature map, and is expressed as:(3)$$P\text{\_}\mathrm{Conv}{\left(w,x\right)}_{\left(i,j\right)}=\sum_{k}^{K}w_{k}•x_{\left(i,j\right)}$$

The fully connected layer structure of DSCNN is replaced with GAP with adaptable dimensions in order to further the goal of quick defect diagnostics of bearings. Specifically, the average value of each feature map is directly mapped to a class label or an output node after the final pooling in the feature extraction layer. The GAP expression is:(4)$$O_{i}=\frac{1}{y_{1}y_{2}}\sum_{m=1}^{y_{1}}\sum_{n=1}^{y_{2}}Y_{m,n}^{i}$$
where *O**_i_* represents the result of global average pooling of the layer I feature map; $Y_{m,n}^{i}$ is the (*m*, *n*) element of the *i*th feature graph in the convolution layer.

The global average pooling layer converts the extracted features into a one-dimensional vector via numerous deep separable convolution and pooling operations. The softmax classifier recognizes the bearing defect category. The probability of the output $y_{i}$ matching the category label is set as follows if the input consists of *N* different types of signals:(5)$$y_{i}=\frac{\exp\left(w_{i}x+b_{i}\right)}{\sum_{i=1}^{N}\exp\left(w_{i}x+b_{i}\right)},i=1,2,3\cdots ,N$$

The categorical cross-entropy function is used to determine the output’s corresponding loss. The discrepancy between the expected and actual output values is represented by the cross-entropy. The value of the associated cross-entropy increases with the size of the difference. The loss function must be minimized as much as possible to obtain the actual output value as close to the predicted output value as possible. The cross-entropy loss function is expressed as follows:(6)$$L=-\frac{1}{m}\sum_{i=1}^{m}\sum_{j=1}^{n}\exp(w_{i}x+b_{i})\log(y_{i})$$

In the formula: *m* is the number of batch samples, *n* is the class to which the samples belong, *x_ij_* is the output value of the fully connected layer, $y_{i}$ is the actual output value of the sample, and $\exp(w_{i}x+b_{i})$ is the predicted output value.

## 3. DS Evidence Theory

Taking into account that a single acceleration vibration sensor’s vibration signal cannot accurately represent the rolling bearing’s actual running condition, the utilization of multi-sensor information can now more accurately depict the bearing’s running condition thanks to advancements in multi-sensor technology. The vibration signals of the bearing to be diagnosed are gathered from various dimensions in accordance with the real configuration of the experimental apparatus in this work, and the diagnostic experiments are conducted accordingly. The diagnostic results of each sensor are fused using the DS evidence theory, and the combined diagnostic results are output once more.

The DS evidence theory, a style of reasoning based on mathematical uncertainty, was created and improved by Dempster and Shafer [29]. Multiple pieces of evidence are used to characterize an issue in the DS evidence theory, and these pieces of evidence are then combined in accordance with predetermined principles to arrive at the final conclusion. Some contradicting and unnecessary information can be removed using this fusion technique. The DS evidence theory must create bodies of evidence for each event and provide fusion rules in order to combine numerous pieces of evidence. Finally, the foundation for decision making is fusion evidence.

Making final decisions is DS evidence theory’s main objective. Therefore, it is necessary to introduce a finite non-empty set as the recognition framework, which contains a variety of events. Each event is independent and in a relatively exclusive relationship. Basic Probability Assignment (BPA) is assigned to various events and is expressed as a function: $m:2^{\theta }\to \left[0,1\right]$ which is satisfied as follows:(7)$$\left\{ \begin{array}{l} \sum_{A\in \Theta }m(X)=1\\ m(\emptyset )=0 \end{array}\right.$$

The function assigns each event a probability representing the level of trust in the event, which represents the degree to which the evidence supports the event, where *m* () represents the uncertainty of the evidence. A confidence interval is introduced to more accurately reflect the level of trust in occurrences. It is made up of two functions, Belief (Bel) and Plausibility (Pl), and has the following expression:(8)$$\left\{ \begin{array}{l} Bel(A)=\sum_{B\subseteq A}m(B)\\ Pl(A)=\sum_{B\cap A\neq \emptyset }m(B) \end{array}\right.$$

The function of trust *Bel*(*A*) reflects the level of support for the event *A*, and *Pl*(*A*), which measures the level of acceptability for the event *A*, represents the level of uncertainty. Figure 5 depicts their relationship to one another.

The application of DS evidence theory in fusion is realized through different levels of trust for the same event. $m_{j}\left(A_{i}\right)$ represents the trust function of the event $A_{i}$ to obtain evidence *i*.
(9)$$M_{final}(A)=\frac{\sum B\cap A_{i}=A\prod_{j=1}^{n}m_{j}(A_{i})}{1-\sum B\cap A_{i}=\emptyset \prod_{j=1}^{n}m_{j}(A_{i})}$$

The fusion order and result of various pieces of evidence are unrelated. The final evidence following fusion is $M_{final}\left(A\right)$. The decision principle allows for the acceptance of the new evidence. The maximum value in final evidence $M_{final}$ is $M_{final}\left(max\right)$, and $M_{final}\left(max\right)$ is at least $\lambda_{1}$ greater than the other values. While $M_{final}\left(max\right)$ and $M_{final}\left(uncertain\right)$ differ by more than $\lambda_{2}$, $M_{final}\left(uncertain\right)$ differs by less than $\lambda_{1}$. The three average decision thresholds in DS evidence theory are $\lambda_{1}$, $\lambda_{2}$, and $\lambda_{3}$, respectively. These three values will be established in the actual reference in accordance with the actual reference scenario.
(10)$$\left\{ \begin{array}{l} M_{final}(\max)>other+\lambda_{1}\\ M_{final}(\max)\geq M_{final}(uncertain)+\lambda_{2}\\ M_{final}(uncertain)<\lambda_{3} \end{array}\right.$$

## 4. Relevant Experiments and Results Analysis

The experimental platform of this research equipment consists of two parts: cloud computing and edge computing. The computing side of the cloud is primarily powered by AMD R7-5800H@4.4MHZ, RAM: 16G, and mostly using the public data set and algorithm library, after which the universal model is loaded and trained. Three machines with the same configuration environment and AMD Ryzen 5 3600 6-Core Processor@3.6GHz, RAM: 16G were used for the edge end. The main work of the edge terminal is to transform the real-time signals collected from the device terminal into trainable sample data, and to load the transformed personalized samples on the universal model unloaded from the cloud to further fine-tune the model weight arranged on the edge node. The Win10 × 64-bit operating system was selected for both the cloud and the edge, a deep learning framework was built based on Python, and the implementation of the entire model training was completed using this framework.

### 4.1. Cloud Diagnostic Algorithm

Case Western Reserve University’s electrical engineering laboratory open motor experimental data collection (http://csegroups.case.edu/bearingdatacenter/home accessed on 1 March 2022) [30] provided the bearing fault data used in this paper’s cloud. The power source is a 2-hp three-phase asynchronous motor, and the acceleration vibration sensor was utilized to record vibration data at the motor fan and drive ends, respectively. At the driving end, an SKF 6205 bearing was utilized, while an SKF 6203 bearing was used at the fan end. In Figure 6, the test platform is displayed.

In order to replicate various pitting flaws frequently seen in actual operation, single pits of varying sizes were consecutively carved on the inner and outer race of the bearing and the rolling body using electric discharge machining (EDM) technology. The size of each flaw was 0.2 mm, 0.3 mm, and 0.5 mm, respectively. The damage sites of the bearing outer ring were among them and were positioned at the three, six, and twelve o’clock positions, respectively. Table 1 displays a cloud-based experimental data collection.

When the training samples are too little, deep learning models frequently overfit. Enhancing the data set is required to fulfill the needs of the quantity of training in order to obtain more efficient training data. This study uses random sampling as a method of data improvement, which not only improves the correlation between sample area information and training samples, but also lowers the likelihood that the model would become overfitted. The length of the chosen vibration data serves as the training sample, and by choosing an acceptable sample length, one could also control the training accuracy and convergence time. The number of fault spots the sensor detected in each rotation of the rotating shaft was between 400 and 416 ((12,000 × 60/1797) ≈ 400), taking into account the bearing speed range of 1730 to 1797 r/min and the sampling frequency of 12 kHz. The number of sampling points for each type of fault was set to 1024 in order to guarantee the reliability of fault data.

This article proposes a DSCNN–GAP (Depthwise Separable Convolution combined with Global Average Pooling, DSCNN–GAP) algorithm. Regarding the source dataset, Figure 7 displays the accuracy and loss of the DSCNN–GAP algorithm. It is evident that the diagnostic outcomes produced by the deeply separable convolution and global average pooling combination strategy are generally favorable. After loading 10 training rounds, the improved DSCNN showed convergence phenomenon in terms of convergence speed, and the accuracy rate on the test set was 95.46%. When the total number of training rounds reached 10~30, the test accuracy varied slightly, but after it reached 30, it stabilized. The model’s failure recognition rate is currently 99.89%, and the loss value has dropped to 0.007. This is due to the fact that DSCNN has improved spatial convolution capabilities and can successfully extract fault features. Additionally, the incorporation of GAP can decrease the number of model parameters, shorten model training, and enhance model robustness to a certain amount. It can be concluded that the DSCNN–GAP algorithm can successfully extract fault features and successfully prevent the occurrence of test set overfitting based on the training speed and diagnostic accuracy.

The experiment compares the suggested approach with the existing mainstream CNN, CNN-GAP, WDCNN [31], and DSCNN algorithms in order to further demonstrate the superiority of the intelligent diagnosis of DSCNN–GAP algorithm described in this research. The sequential input of the aforementioned comparison method is still the data set displayed in Table 1 for testing purposes. The average training time, the average diagnostic accuracy, and the average loss accuracy are all calculated from the experimental data of 10 repeats to ensure the experiment’s validity. Table 2 displays the initial parameters and diagnostic outcomes for the five algorithms.

The table makes it very evident that the diagnostic accuracy of CNN, CNN-GAP, WDCNN, and DSCNN was 95.12%, followed by 95.75%, 89.07%, and 97.99%. DSCNN–GAP achieved a diagnostic accuracy of 99.89%, which is much higher than previous comparable methods. The proposed algorithm’s diagnostic accuracy and loss are clearly superior to those of WDCNN, despite the fact that its average training time is slightly slower than that of WDCNN. It is important to note that the proposed DSCNN–GAP model uses an approach that combines deeply separable convolution with global average pooling, and because there are fewer model parameters than other comparison algorithms, the model training time is dramatically reduced. According to the experimental findings, the suggested DSCNN–GAP algorithm outperforms other diagnostic algorithms.

### 4.2. Effect Analysis of Edge End before and after Transfer Experiment

The CUT-2 experimental platform gathered some data at the edge that is comparable to the fault in the cloud data set. A deep groove ball bearing with model 6900 ZZ was chosen for this experiment. EDM technology pitted the defective bearing. The fault diameters were 0.2 mm and 0.3 mm, and they were dispersed across the bearing’s inner race, outer race, and the ball. When the rotating shaft speed was 2000 r/min, the vibration signals of the defective bearing in the *X*, *Y*, and *Z*-axis directions were gathered in accordance with the various distribution directions of the acceleration vibration sensors. The data acquisition card’s sample frequency was set to 5 k, and the sensor collected 150 (5000 × 60/2000 ≈ 150) points per shaft revolution. The chosen sample length was 512 to guarantee the accuracy of the fault data. Table 3 displays the chosen rotating bearing defect data sets for the X, Y, and Z axes.

The direct application of the cloud DSCNN–GAP pre-training model to the edge side may not produce the best results due to variances in the training data of the participating cloud-side models. This paper introduces transfer learning knowledge in order to improve performance [14,32]. Transfer learning is seen to have a significant deal of promise for completing varying but related activities from the source domain to the target domain. The most popular transfer learning technique is parameter transfer, which aims to offer useful parameter knowledge for the target model from a good pre-training model (source model). The specific operation is the method of training some layers (the retraining layer) while first freezing the parameters of other layers (the frozen layer) in the pre-training model, which is a practical technique to apply the pre-training model to different scenarios. In the structure of the deep learning model, the earlier layers contain more general functions, while the later layers gradually become focused on class details. Here, all feature extraction layers are frozen, and the classification and complete connection layers are adjusted to the target state. Figure 8 depicts the suggested transfer learning technique.

#### 4.2.1. Experimental Results of No-Transfer Learning at the Edge End

The same device adopted multisensory technology, meaning that pertinent sample data were made in accordance with Table 3 for the data acquired along the *X*, *Y*, and *Z* axes, and were fed to the associated edge nodes in turn for diagnosis. The experimental results are displayed in Table 4, Table 5 and Table 6.

Table 4, Table 5 and Table 6 display the diagnostic accuracy of various fault data obtained by the X, Y, and *Z*-axis vibration sensors under various sample sizes and training iterations without migration. First of all, we can observe that there are variations in the diagnostic accuracy of bearings since the signals gathered by vibration sensors in various directions represent the operating state of bearings. In particular, as sample size and training rounds increased, the average diagnostic accuracy eventually became close to or achieved 1. For instance, when the sample size was 50, after adequate training, edge nodes organized in the *X*-axis direction had superior diagnostic accuracy than vibration sensors arranged in the Y and Z directions, which were, respectively, 1.0, 0.8857, and 0.9286.

#### 4.2.2. Experimenting with Transfer Learning at the Edge

Firstly, the ideal model weight was preserved once the DSCNN–GAP algorithm had been fully trained using the open fault bearing data set in the cloud. Figure 9 illustrates that when the ideal model weight is kept, the defect diagnostic accuracy can reach 99.89%. The weight of the model was unloaded to each edge node, and then the transfer learning experiment was carried out and the edge nodes positioned on the *X*, *Y*, and *Z*-axis vibration sensors were each put to the test in turn. The outcomes are displayed in Table 7, Table 8 and Table 9.

The findings of the migration experiment on the *X*-axis are provided in Table 7 as a diagnosis. The diagnostic accuracy after migration has often greatly increased when compared to the findings of the experiment without migration on the *X*-axis. It is noteworthy that the diagnostic accuracy can reach 92.86% when there are 20 samples in each category and 30 training rounds, whereas the diagnostic accuracy in the absence of the migration experiment is only 71.43% under the same conditions. Similar to this, the diagnostic accuracy is as high as 92.86% when each type of fault sample is 50 and there are only 5 training rounds, but under the same conditions, the diagnostic accuracy without migration experiment is only 32.86%. From the data above, it is clear that the *X*-axis transfer experiment’s diagnostic effect has greatly improved and that a better diagnostic effect may be attained despite the dual restrictions of a small number of training samples and training rounds.

The outcome of the migration experiment on the *Y*-axis is displayed in Table 8. The diagnostic accuracy after migration is likewise much better than the outcome of the experiment without migration on the *Y*-axis. When there are 20 samples in each category and the diagnostic accuracy is fully trained, the *Y*-axis transfer experiment’s diagnostic accuracy can reach 92.86%, whereas the diagnostic accuracy without the experiment is only 85.71% under the same circumstances. With the same number of samples, the *X*-axis data transfer experiment would require 10 training rounds to reach the same diagnostic accuracy. In a similar vein, the diagnostic accuracy in the *Y*-axis migration experiment is as high as 91.43% when each type of fault sample is 50 and the number of training rounds is just 10, compared to only 70% in the case without the migration experiment. Only 5 training rounds were required to reach a superior diagnostic accuracy of 92.86% in the *X*-axis transfer diagnostic trial. As can be seen from the analysis above, even if the *Y*-axis produces a decent diagnostic result when performing a migration experiment, when the sample size is the same, the *X*-axis can produce a superior diagnostic result by quickly fine-tuning the parameters.

The diagnostic outcomes of the *Z*-axis migration experiment are displayed in Table 9. In general, both before and after the migration, the *Z*-axis was weaker than the *X*-axis and *Y*-axis. The diagnostic impact of the data gathered on the *X*-axis was the best before and after migration, according to a summary of the experimental results before (Table 4, Table 5 and Table 6) and after (Table 7, Table 8 and Table 9) migration. This suggests that both the sensor’s position and the bearing’s operational condition are different. Because the bearing rotor experiment platform is vertically oriented when collecting vibration signals, this makes the vertical direction vibration sensor response by the running state of the bearing the most sensitive. This offers the suggestion to put sensors along the vertical axis, which would be the least expensive and would best reflect the equipment’s operational state.

#### 4.2.3. Experimental Results of Multi-Edge Node Decision Fusion

The vibration signals gathered by the vibration sensors on the *X*, *Y*, and *Z*-axis have substantially enhanced the diagnostic accuracy of the experiment after migration, as can be observed from the above diagnostic results. The organization of numerous sensors in various directions can precisely address this issue, as the fault data gathered by a single dimensional sensor cannot adequately reflect the running state of rolling bearings. In order to increase the accuracy of the diagnosis, this paper employs the multi-classifier decision fusion method. First, the DS evidence fusion algorithm is used to make decision fusion for the classifiers in three directions under each circumstance based on the aforementioned diagnostic results of the single-direction vibration sensor migration experiment.

Table 10 and Figure 9 illustrate the accuracy of the bearing diagnosis following the fusion of the defect data from vibration sensors oriented differently using the DS evidence theory. It is important to note that when only 10 samples and 10 training rounds were used, a diagnostic accuracy of more than 94.38% could be attained. The highest diagnostic result of a single sensor after migration was only 78.57%. The model after multi-edge node decision fusion achieved 99.05% diagnostic accuracy when the training sample count was 200 only by fine-tuning the training parameters, while the best diagnostic accuracy of a single sensor was 82.14%. The reason for this is that the signal fusion of differently-positioned vibration sensors is successful, because the vibration fault characteristics of the various position vibration sensors were combined using the DS evidence theory. This allowed for a more thorough reflection of the running state of the bearings and, ultimately, produced good diagnostic results. In conclusion, because the actual factory environment is much more complex than the experimental conditions, it is difficult to determine which direction of the sensor can accurately diagnose the equipment’s running status. In this paper, the migration after learning of the bearing fault diagnosis results fusion was used to address this issue. It aimed to jointly reflect the running state of bearings by capturing the signal of multi-azimuth sensors.

The experimental results of a small number of samples in relevant experiments before and after migration were respectively intercepted and summarized for comparative analysis. The summarized table is shown in Table 11, and the following conclusions can be made:(1)The accuracy of fault diagnosis can be considerably increased by using transfer learning on small sample data. The diagnostic accuracy of the *X*-axis, *Y*-axis, and *Z*-axis without migration was 71.43%, 71.43%, and 50%, respectively, when the training samples for each category were only 10, and the average diagnostic accuracy was 64.29%. However, the edge nodes’ accuracy of diagnosis after migration was 85.71%, 85.71%, and 78.57%; the average diagnostic accuracy was 83.33%. As can be observed, the accuracy of the diagnosis after migration was 19.04% greater than that of the diagnosis prior to migration. The highest diagnostic accuracy of a single edge node after transfer learning was 96.43%, which is 12.51% higher than that in the unmigrated state, when training samples of each category were increased to 20. The highest diagnostic accuracy of each category, however, was 85.71% in the unmigrated state.(2)The training period can be drastically cut down using the transfer learning method. The average training time of various sample counts on the *X*-, *Y*-, and *Z*-axis is presented here in order to reduce experimental error while taking into account the consistency of the hardware equipment of each edge node. When transfer learning was not used, the fault data was imported directly with training sample sizes of 10 and 20. The times were 2.7031 s and 3.8712 s, respectively, after 40 training rounds, while the greatest diagnostic accuracy was only 85.71%. The model’s training took 2.1889 and 3.0531 s in the migration state, respectively, and its best diagnostic precision was 96.43%. As can be shown, the speed of transfer learning-based model fine-tuning is slower than that of a new training model in the case of tiny sample data, but the diagnostic outcome is better.(3)After migration, the DS evidence theory’s diagnostic precision greatly increased. The diagnostic effect of transfer learning greatly improves the recognition rate when there are few training examples, yet it still exhibits instability. Multi-sensor fusion technology can more precisely identify bearing fault categories.

### 4.3. Time Analysis of Cloud—Edge Collaborative Diagnosis

The time of cloud-side collaborative diagnosis must be further examined in light of the aforementioned conclusions in order to confirm that the cloud-side collaboration suggested in this study may speed up model training speed and accomplish quick diagnosis. The following limiting criteria are suggested in order to verify the validity of the experimental results and minimize the interfering variables impacting the results:(1)Special problems in network communication, such as packet loss rate, are not considered.(2)When communicating, data compression technology (calculation in professional communication) is not considered.

Sampling time is $T_{t1}$, $T_{t2}$ represents the data upload time from the edge layer to the cloud layer, $T_{train\_c}$ represents the training time of the model in the cloud, and $T_{diag\_c}$ represents the testing time of the model in the cloud. $T_{t3}$ represents the transmission time from cloud layer to edge layer. $T_{train\_e}$ represents the training time of the model in the edge end, and $T_{diag\_e}$ represents the testing time of the model in the edge end.

The amount of time needed to gather the sample size that the cloud requires is referred to as the data sampling time. $T_{t1}$ is the time required to collect *X* points in sample *M*, the frequency is *f*, and the data sampling time is as follows:(11)$$T_{t1}=\frac{MX}{f}$$

The amount of time needed to send data from the edge to the cloud is shown by the data upload time $T_{t1}$. This section is heavily impacted by network fluctuations. Shannon’s theorem [33] states that the quickest way to send data of size *C* over a network channel with bandwidth of size *B* is as follows:(12)$$T_{t2}=C/(Blb(1+S/N))$$
where *S* is the average signal power, *N* is the average noise power, and *S*/*N* is signal-to-noise ratio.

The total time of diagnosis using the cloud–edge collaborative framework is:(13)$$T_{total}=T_{t1}+T_{t3}+T_{train\_e}+T_{diag\_e}$$

The total time of diagnosis using the cloud computing framework is:(14)$$T_{total}=T_{t1}+T_{t2}+T_{train\_c}+T_{diag\_c}$$

The diagnostic tests conducted by each edge node under the transfer learning scenario demonstrate that good diagnostic outcomes can be obtained with a modest sample size and few training cycles. From the aforementioned diagnostic results, it can be seen that when the number of samples in each category is set to 50, the diagnostic accuracy of edge nodes on the *X*-, *Y*-, and *Z*-axis is 98.57%, 97.14% and 95.86%, respectively, after full fine-tuning, and the average diagnostic accuracy is 97.19%, which can satisfy the diagnostic objective. The aforementioned non-transfer learning experiment shows that when 95% of the diagnostic accuracy is achieved, each edge node should have at least 100 training samples. Therefore, the time needed for diagnosis using cloud–edge collaboration and cloud computing frameworks was calculated, both under the assumption of obtaining a same level of diagnostic accuracy.

Table 12 shows that, in the case of greater cloud computing resources, the training time needed by the cloud is 76.143 s, whereas the training and diagnostic time needed by the edge utilizing transfer learning is just 8.219 s, or roughly 1/9 of the time required by the cloud. The overall diagnostic time for cloud computing is 97.647 s, while the total diagnostic time for cloud–edge collaboration is 18.971 s, or around 1/5 of the latter, assuming that the two methods achieve comparable diagnostic accuracy. For the same diagnostic accuracy as cloud computing, cloud–edge collaboration requires fewer samples and a shorter sampling period. A significant amount of feature extraction time can be saved by using the transfer learning method. The use of GAP rather than a full connection layer can significantly reduce the amount of model parameters, allowing for the speedy fine-tuning of a customized diagnostic model appropriate for edge nodes even while the output layer still needs to be retrained.

## 5. Conclusions and Future Work

This paper proposes an improved DSCNN–GAP lightweight method to conduct diagnostic research on a cloud–edge collaborative architecture for real-time diagnosis of bearing problems. First, the cloud model’s training was carried out using the improved DSCNN–GAP algorithm and the common bearing fault data set. The cloud model was continuously updated to be more appropriate for edge-end diagnosis with the uploading of individualized fault data at each edge node and the expansion of the cloud database. The cloud initial training model was then downloaded to each edge node. A small number of personalized fault samples gathered at the edge node were then used to quickly fine-tune the universal model’s parameters, and the customized model created could then quickly perform diagnostic experiments on the source data. Additionally, it is frequently impossible for a single acceleration vibration sensor to fully monitor the running condition of rolling bearings. In order to make decision fusion for the diagnostic findings at each edge node, multi-sensor technology was therefore introduced in this research. DS evidence theory was then employed to do so, and the fusion results further increase the reliability of diagnostic results. The bearing fault diagnostic method can be used to finish training the diagnostic model in the cloud through experimental verification and comparative analysis, while the edge end only needs to make minor adjustments to the universal model that was downloaded from the cloud in order to take part in the diagnosis. Model training time can be significantly reduced with this cloud–edge collaborative diagnostic approach. At the same time, the edge only needs to collect a small number of personalized samples to meet the needs of diagnosis, saving a lot of sample data collection and labeling time. The experimental findings demonstrate that the strategy suggested in this paper has a positive impact on the bearing defect diagnosis’s real-time performance, accuracy, and sample limitation. Moreover, it offers a fresh concept and a broad framework for defect diagnostics that may be easily applied to mechanical and industrial systems.

Unsupervised migration learning situations are not covered in this study; only the fault diagnostic scenarios with supervised migration learning have been covered. In the subsequent study, a feature network is constructed for the fault data between the source domain and destination domain from the standpoint of feature correlation in order to create a fault diagnostic scenario that is more appropriate.

## Figures and Tables

**Figure 1 entropy-24-01277-f001:**
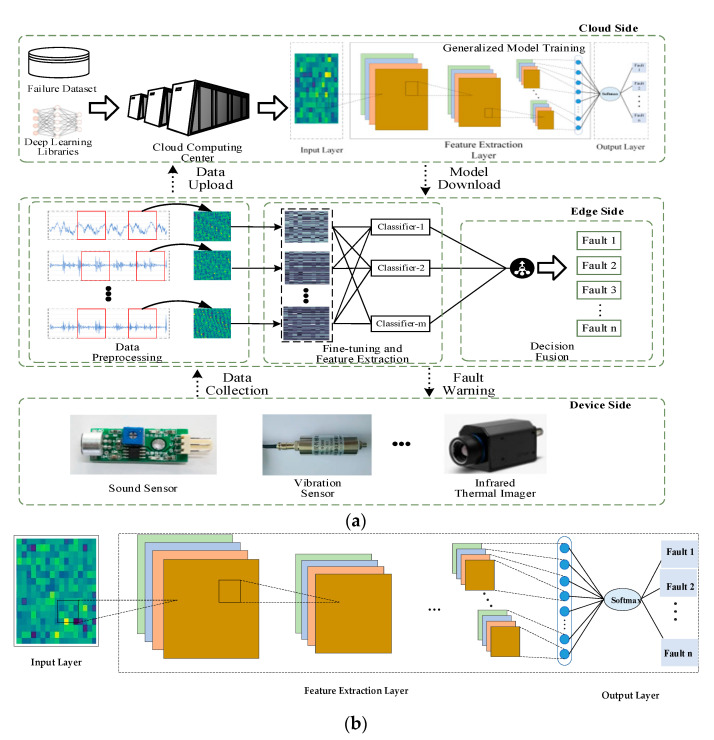
(**a**) Description of Fault bearing diagnostic framework based on cloud and edge collaboration. (**b**) Description of a larger view of Generalized Model Training in the cloud side.

**Figure 2 entropy-24-01277-f002:**
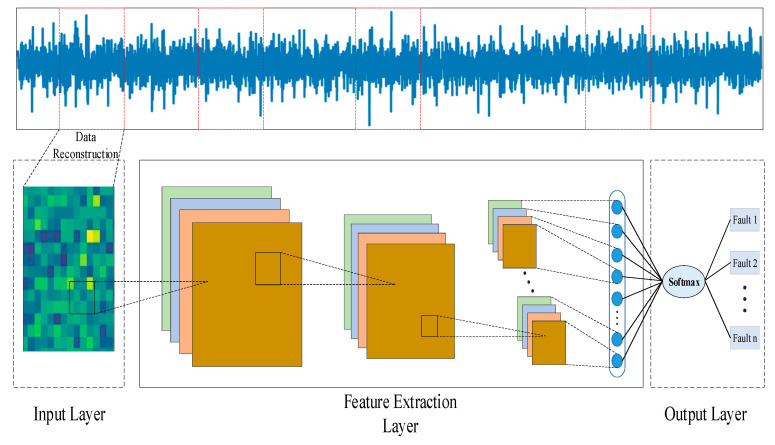
DSCNN–GAP diagnostic algorithm.

**Figure 3 entropy-24-01277-f003:**
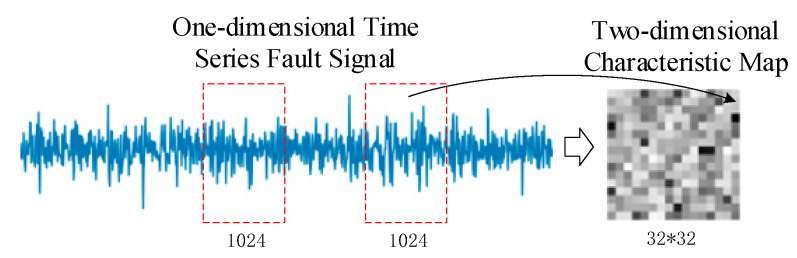
Schematic diagram of data reconstruction.

**Figure 4 entropy-24-01277-f004:**
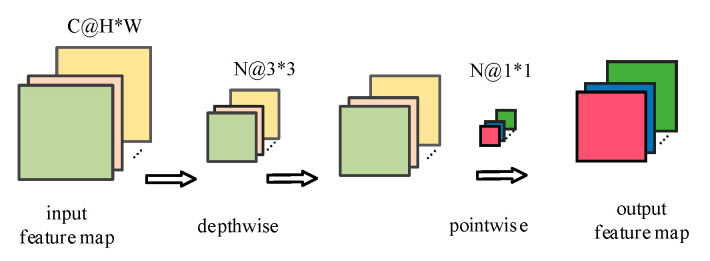
Schematic diagram of depth-separable convolution feature extraction.

**Figure 5 entropy-24-01277-f005:**
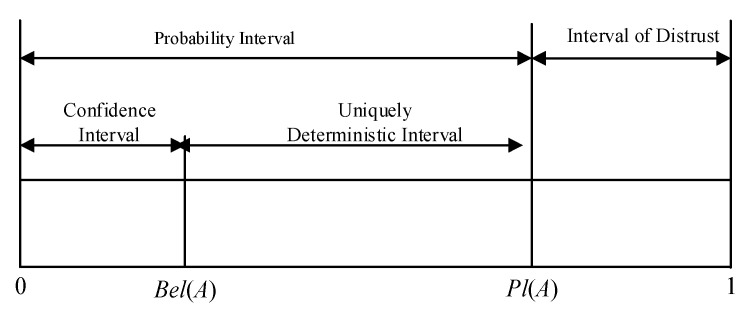
Uncertain description of proposition *A*.

**Figure 6 entropy-24-01277-f006:**
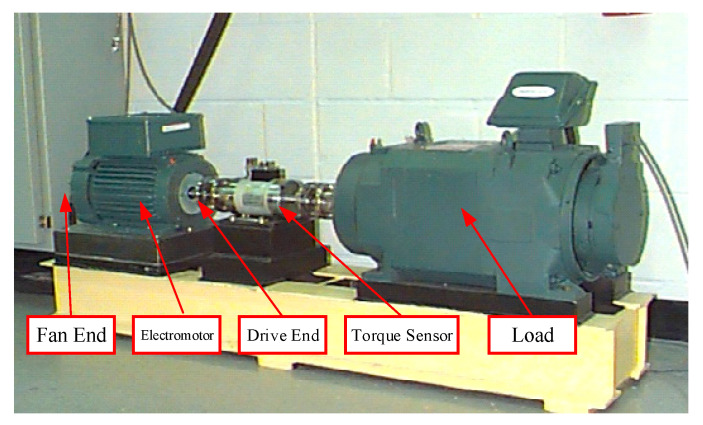
CWRU rolling bearing data acquisition test rig.

**Figure 7 entropy-24-01277-f007:**
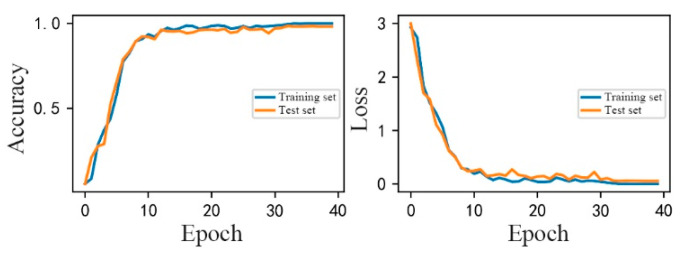
Diagnostic effect of DSCNN–GAP algorithm in the cloud.

**Figure 8 entropy-24-01277-f008:**
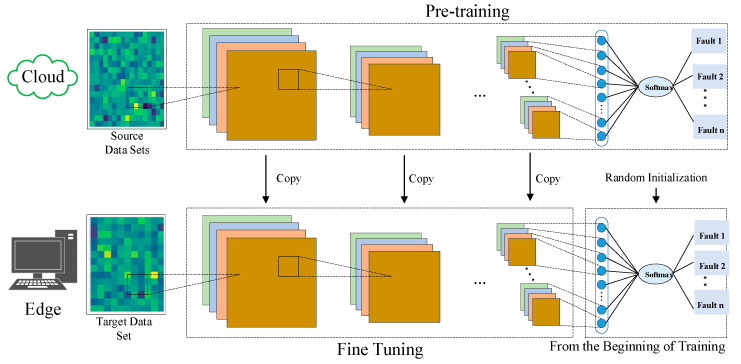
Schematic diagram of the transfer learning method.

**Figure 9 entropy-24-01277-f009:**
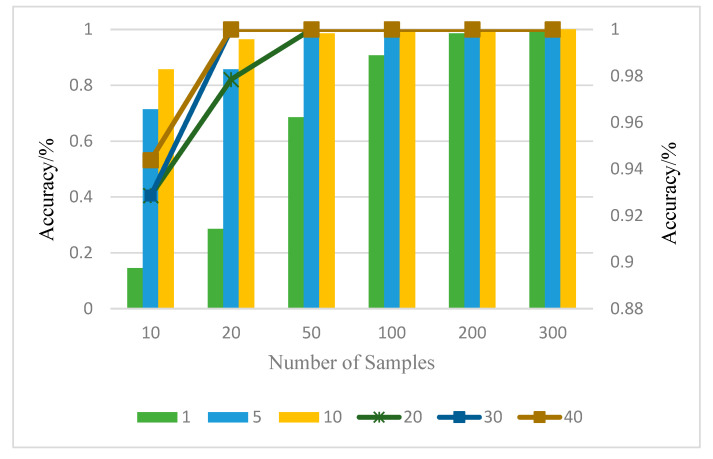
Results of DS decision fusion.

**Table 1 entropy-24-01277-t001:** Cloud experimental sample dataset.

Sensor Installation Position	the Fault Size	Fault Location	Label
Normal bearing			0
Drive End	0.2 mm	inner race	1
		the ball	2
		outer race X/Y/Z	3
	0.3 mm	inner race	4
		the ball	5
		outer race X/Y/Z	6
	0.5 mm	inner race	7
		the ball	8
		outer race X/Y/Z	9
Fan End	0.2 mm	inner race	10
		the ball	11
		outer race X/Y/Z	12
	0.3 mm	inner race	13
		the ball	14
		outer race X/Y/Z	15
	0.5 mm	inner race	16
		the ball	17
		outer race X/Y/Z	18

**Table 2 entropy-24-01277-t002:** Comparison of diagnostic accuracy of different diagnostic models in CRWU dataset.

Diagnostic Model	Depth of Network	Learning Rate	Number of Parameters	Average Training Time	Average Accuracy	Average Loss
CNN	3	0.005	668787	47.2735	0.9512	0.1949
CNN-GAP	3	0.005	126451	36.9851	0.9575	0.1797
WDCNN	5	0.01	42419	61.9839	0.8907	0.6077
DSCNN	3	0.005	188691	43.4858	0.9799	0.0719
DSCNN–GAP	3	0.005	39571	30.8380	0.9989	0.0071

**Table 3 entropy-24-01277-t003:** Edge end experimental data set.

Axial Direction	Fault Location and Diameter			Sample Length	Sample Label
	Outer Race Fault	Inner Race Fault	Ball Fault		
	Diameter/mm	Diameter/mm	Diameter/mm		
X	Normal			512	0
	0.2	Null	Null	512	1
	Null	0.2	Null	512	2
	Null	Null	0.2	512	3
	0.3	Null	Null	512	4
	Null	0.3	Null	512	5
	Null	Null	0.3	512	6
Y	Normal			512	0
	0.2	Null	Null	512	1
	Null	0.2	Null	512	2
	Null	Null	0.2	512	3
	0.3	Null	Null	512	4
	Null	0.3	Null	512	5
	Null	Null	0.3	512	6
Z	Normal			512	0
	0.2	Null	Null	512	1
	Null	0.2	Null	512	2
	Null	Null	0.2	512	3
	0.3	Null	Null	512	4
	Null	0.3	Null	512	5
	Null	Null	0.3	512	6

**Table 4 entropy-24-01277-t004:** Experimental results of *X*-axis without migration.

Number of Samples	Epoch					
	1	5	10	20	30	40
10	0.1429	0.1429	0.2857	0.4286	0.5	0.7143
20	0.1429	0.1429	0.3571	0.6786	0.7143	0.8214
50	0.1429	0.3286	0.7286	0.9857	0.9714	1.0
100	0.1643	0.6929	0.8214	0.9571	0.9786	0.9929
200	0.2964	0.8464	0.9750	0.9893	0.9893	0.9929
300	0.2857	0.9810	0.9952	0.9929	0.9929	0.9976

**Table 5 entropy-24-01277-t005:** Experimental results of *Y*-axis without migration.

Number of Samples	Epoch					
	1	5	10	20	30	40
10	0.1429	0.1429	0.3571	0.4286	0.6429	0.7143
20	0.1429	0.2143	0.2857	0.7143	0.8214	0.8571
50	0.1429	0.4	0.7	0.9286	0.8857	0.8857
100	0.1429	0.8643	0.8571	0.9643	0.9786	0.9929
200	0.1429	0.9643	0.9857	0.9929	0.9929	0.9964
300	0.2881	0.7738	1.0000	0.9976	1.0	1.0

**Table 6 entropy-24-01277-t006:** Experimental results of *Z*-axis without migration.

Number of Samples	Epoch					
	1	5	10	20	30	40
10	0.1429	0.1429	0.1429	0.2857	0.3571	0.5000
20	0.1429	0.1429	0.25	0.4643	0.4643	0.5357
50	0.1429	0.1429	0.4429	0.6857	0.8857	0.9286
100	0.1429	0.4857	0.6286	0.7000	0.9286	0.9786
200	0.1429	0.5929	0.7821	0.9786	0.9893	0.9893
300	0.2833	0.8310	0.9786	1.0	0.9976	1.0

**Table 7 entropy-24-01277-t007:** *X*-axis migration experiment results.

Number of Samples	Epoch					
	1	5	10	20	30	40
10	0.1786	0.3571	0.7143	0.7857	0.7857	0.8571
20	0.2143	0.5714	0.8214	0.8929	0.9286	0.9643
50	0.5857	0.9286	0.8571	0.9429	0.9714	0.9857
100	0.6929	0.9786	0.9643	0.9786	0.9929	1.0
200	0.8214	0.9786	0.9821	0.9929	0.9929	0.9964
300	0.8667	0.9929	0.9976	0.9952	1.0	1.0

**Table 8 entropy-24-01277-t008:** *Y*-axis migration experiment results.

Number of Samples	Epoch					
	1	5	10	20	30	40
10	0.1429	0.5714	0.5	0.6429	0.7857	0.8571
20	0.2857	0.75	0.8214	0.8929	0.8929	0.9286
50	0.4143	0.8429	0.9143	0.9286	0.9571	0.9714
100	0.6429	0.9286	0.9286	0.9643	0.9786	0.9857
200	0.8179	0.9536	0.9714	0.9786	0.9929	0.9964
300	0.8810	0.9738	0.9833	0.9905	0.9952	1.0

**Table 9 entropy-24-01277-t009:** *Z*-axis migration experiment results.

Number of Samples	Epoch					
	1	5	10	20	30	40
10	0.1429	0.3571	0.5714	0.5714	0.7143	0.7857
20	0.2857	0.5714	0.6786	0.7143	0.75	0.8214
50	0.4143	0.7429	0.8143	0.8286	0.9143	0.9286
100	0.6786	0.8429	0.8857	0.9857	0.9357	0.9929
200	0.7536	0.9964	0.9964	1.0	0.9964	1.0
300	0.8095	0.9048	0.9952	1.0	0.9976	1.0

**Table 10 entropy-24-01277-t010:** Experimental results of DS decision fusion.

Number of Samples	Epoch					
	1	5	10	20	30	40
10	0.1448	0.7143	0.8571	0.9286	0.9286	0.9438
20	0.2857	0.8571	0.9643	0.9785	1.0	1.0
50	0.6857	0.9857	0.9857	1.0	1.0	1.0
100	0.9071	1.0	1.0	1.0	1.0	1.0
200	0.9857	1.0	1.0	1.0	1.0	1.0
300	0.9976	1.0	1.0	1.0	1.0	1.0

**Table 11 entropy-24-01277-t011:** Comparison of the effect before and after migration under the condition of few samples.

Adopt Strategy	Epoch					
	1	5	10	20	30	40
Unmigrated10-X	0.1429	0.1429	0.2857	0.4286	0.5	0.7143
Unmigrated 20-X	0.1429	0.1429	0.3571	0.6786	0.7143	0.8214
Unmigrated 10-Y	0.1429	0.1429	0.3571	0.4286	0.6429	0.7143
Unmigrated 20-Y	0.1429	0.2143	0.2857	0.7143	0.8214	0.8571
Unmigrated 10-Z	0.1429	0.1429	0.1429	0.2857	0.3571	0.5000
Unmigrated 20-Z	0.1429	0.1429	0.25	0.4643	0.4643	0.5357
Fine-tuning10-X	0.1786	0.3571	0.7143	0.7857	0.7857	0.8571
Fine-tuning20-X	0.2143	0.5714	0.8214	0.8929	0.9286	0.9643
Fine-tuning10-Y	0.1429	0.5	0.5714	0.6429	0.7857	0.8571
Fine-tuning20-Y	0.2857	0.75	0.8214	0.8929	0.8929	0.9286
Fine-tuning10-Z	0.1429	0.2857	0.3571	0.5714	0.7143	0.7857
Fine-tuning20-Z	0.2857	0.5714	0.6786	0.7143	0.75	0.8214
DS-10	0.1448	0.7143	0.8571	0.9286	0.9286	0.9438
DS-20	0.2857	0.8571	0.9643	0.9785	1.0	1.0

**Table 12 entropy-24-01277-t012:** Comparison of diagnostic time under different frames.

Frames	*T_t_* _1_	*T_t_*_3_/*T_t_*_3_	*T_train_e_ *+ *T_diag_e_*/*T_train_C_ *+ *T_diag_C_*	*T_total_*
Cloud Edge Collaboration	10.24 s	0.512 s	8.219 s	18.971 s
Cloud Computing	20.48 s	1.024 s	76.143 s	97.647 s

## Data Availability

The data presented in this study are openly available in Case Western Reserve University’s electrical engineering laboratory open motor experimental data collection (https://engineering.case.edu/bearingdatacenter, accessed on 1 March 2022).

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
