# Peer review of "Research on the Rapid Diagnostic Method of Rolling Bearing Fault Based on Cloud–Edge Collaboration"

_entropy, 2022, doi:10.3390/e24091277_

Round 1

Reviewer 1 Report

I would recommend authors use proper units in tables while presenting results. Please mentions what DSCNN-GAP stands for and what is the reason to select this type of technique.

Reviewer 2 Report

The authors propose a DSCNN-GAP lightweight method to monitor bearing faults on a cloud-edge collaborative architecture for real-time diagnosis of bearing problems.

 a) This reviewer is not sure if the author’s statement: “The majority of the mechanical fault diagnosis techniques currently in use are based on cloud computing models” is completely true. This reviewer has read several articles about multiple faults in an induction motor and several techniques use the monitoring of current of vibration and the application of some methodology. Please, the authors should maybe modify this statement. I think that recently some methodologies use the cloud computing models, but the traditional acquisition of signals is still being used with the application of some new methods.

 b) Could the authors state what are the advantage of using of cloud computing models over traditional acquisition and signal processing techniques?.

c) Some of the traditional techniques are capable of do real-time fault identification, and this reviewer is not sure if the models based on cloud computing can be able to do real-time monitoring, checking table 12, it seems that diagnostic time is high to do real-time monitoring. So, there is some possibility this time can be reduced, and a real time-time monitoring can be used.

d) Does Some other types of faults can be identified by using the proposed methodology?

Reviewer 3 Report

This paper dealt with a widely treated issue. Indeed, the literature is rich of proposal ranging from signal processing- to machine learning-based techniques to address the issue induction motor fault diagnosis.

The paper is of interest, but unfortunately, the proposed paper suffers from some lacks acting against its consideration:

The state-of-the-art review has been poorly conducted and is based on a reference section missing many important and relevant papers. In this context, the introduction section, even long, is inconsistant.

Round 2

Reviewer 3 Report

Thank you to authors for the new version.